# Alignment of the Sternum and Sacrum as a Marker of Sitting Body Posture in Children

**DOI:** 10.3390/ijerph192316287

**Published:** 2022-12-05

**Authors:** Wojciech Piotr Kiebzak, Arkadiusz Łukasz Żurawski, Michał Kosztołowicz

**Affiliations:** 1Institute of Health Sciences, Collegium Medicum, The Jan Kochanowski University in Kielce, 25-369 Kielce, Poland; 2Świętokrzyskie Centre for Paediatrics, Provincial Integrated Hospital in Kielce, 25-736 Kielce, Poland; 3Kieleckie Towarzystwo Naukowe, 25-303 Kielce, Poland

**Keywords:** body posture, sternum, sacrum, sitting position, kyphosis, lordosis

## Abstract

An analysis of literature on the methods of assuming a sitting position and the results of our own research indicated the need to search for biomechanical parameters and existing relationships that would enable a description of sitting body posture. The purpose of this paper is to analyze the relationship between the alignment of the body of sternum and sacrum and the changes in the thoracic and lumbar spine curvatures in children. The study involved 113 subjects aged 9–13 years. A planned simultaneous measurement of the angle parameters of the alignment of the body of sternum and sacrum relative to the body’s sagittal axis and the angle parameters of the thoracic and lumbar spine curvatures was performed during a single examination session. The proposed markers of alignment in the corrected sitting body posture are characterized by homogeneous results. A high measurement repeatability was observed when determining the corrected body posture in the study setting. It was noted that changes in the alignment of the body of sternum and sacrum resulted in changes in the thoracic kyphosis and lumbar lordosis angle values, which may be an important component of clinical observations of sitting body posture in children. Implementing the body of sternum alignment angle of about 64° relative to the body’s sagittal axis in clinical practice as one of the objectives of postural education may be the target solution for sitting body posture correction in children.

## 1. Introduction

Gradual change in the shape of the spine is an adaptation mechanism to changing conditions and motor needs which accompany a person during their growth [1]. The shape of the spine is formed by many factors, both internal and external, and it changes throughout life, under the influence of age, kind of performed work, but also emotional state and physical fatigue [2,3].

It should be emphasized that disorders in the shape of the spine often occur as a result of bad habits [4]. The aforementioned habits may result in the adoption of a non-physiological body posture in everyday activities, influencing the shaping and preserving an abnormal body shape [4], which consequently negatively affects many body systems [5,6,7,8,9,10,11,12,13].

Some of these disorders, despite apparent regression, reappear later in life, most often as weakening of manual skills and graphic functions, as well as disorders in the development of eye–hand coordination, visual analysis and synthesis [14]. All these factors may lie behind psychosocial problems [15].

Attention is drawn to the need to assess the body posture of young people, especially in the period of rapid growth [2,3,4]. The need to make children aware of the need to control body posture is also indicated [3,4].

Getting into a sitting position is a typical activity of daily living. An analysis of the results of studies on time spent sitting should encourage an in-depth reflection on the manner of sitting. In recent years, time spent sitting has increased significantly [16]; this phenomenon is characterized by the fact that people nowadays spend up to 80% of their working time in a sitting position [17]. Unfortunately, the dynamic development of modern technologies may contribute to an escalation of these behaviors. Long-term clinical observations of the authors show that long-term sitting during the day also applies to children. Moreover, clinical practice shows that children suffer from the same complications associated with static overload of the spine. Young people are less aware of the potential complications of these overloads, and their attitude to correction is marked by a certain nonchalance. The young generation requires clinicians to perform activities that are easy to perform and effective at the same time.

Consideration of the “sedentary lifestyle” and the quality of various sitting postures raises two fundamental questions: whether there is an ideal, most appropriate sitting posture [18,19,20,21] and whether it is beneficial to maintain the proper spinal curvatures while maintaining a proper sitting posture [19].

The authors have assumed that the proper sitting posture is the neutral position, referred to as the “short lordosis posture,” characterized by easy maintenance of the physiological spinal curvatures with a moderate extension of the lumbar region. Maintaining this posture should ensure, among others, a reduced apophyseal joint load as compared to the alignment of spinal segments in the final range of motion, i.e., maximum flexion or extension [19]. It should be emphasized that, contrary to the common opinion that this posture results from the sum of muscle and muscle group forces, it is determined by the integration of visual, vestibular and somatosensory stimuli [22,23,24]. Its quality depends on will to a small degree and results from the correct development and decline of tonic reflexes, among others [25].

It was found that uttering corrective remarks, such as “don’t slouch,” “pull your shoulder blades down,” “sit straight” or “straighten up” does not provide clear guidance [26,27] and does not improve sitting body posture. Actions that consider the physiological parameters of the body, in accordance with human psychophysical abilities, may be considered the right method of shaping the sitting body posture.

An analysis of the literature on these topics and the results of our own research [28] indicated the need to search for biomechanical parameters and existing relationships that would enable a description of sitting body posture. Euclidean geometry [29] helped in solving these tasks, showing that the common relationships between specific body parts, especially the sternum, sacrum, thoracic kyphosis and lumbar lordosis, referred to as the “common sense” [30], may be a method of sitting posture evaluation.

The authors hypothesized that the correction of the posture in a sitting position requires a precise determination of the direction of the correction and its value. Being aware of the common problem of an incorrect sitting position, it was assumed that the tools to support its correction must be widely available and easy to use. A commonly available digital inclinometer was used to determine the parameters of the sternum and sacrum position. The effect of these actions was verified using the aforementioned Euclidean geometry.

### Objective

To analyze the relationship between the alignment of the body of sternum and sacrum and the changes in the thoracic and lumbar spine curvatures in children.

## 2. Materials and Methods

### 2.1. Study Subjects

The study involved 113 subjects, including 43 boys (38.05%) and 70 girls (61.95%). Subject age ranged from 9 to 13 years. Mean height in the study group was 1.41 m (+/−0.20), mean weight was 44.06 kg (+/−9.57) and BMI was 21.92 (+/−1.32).

Thoracic kyphosis takes its shape after the child is 7 years old. Lordosis at this time is already pre-established, but it may still undergo some slight changes during puberty. The authors decided to conduct a study in children of this age to show the biomechanics of the child’s body before the changes in the pubertal period.

Subjects were enrolled into the study randomly. For the experiment, participants were recruited from among children presenting for routine body posture testing. All children meeting the inclusion criteria were selected consecutively until the presented number of qualified participants was reached. Participants were reported for check-ups by their guardians using the IT system.

Inclusion criteria included: (1)good general health—patients with a result of 0 or 1 in the WHO ECOG scale were qualified,(2)no spinal pain during the study and in the three months prior to the audit,(3)normal structure of the chest and spine—this condition was assessed by an experienced physical therapist.

Exclusion criteria included: (1)back pain occurring during the examination and in the three months preceding the examination (the project assumes determining the parameters of the alignment of individual parts of the body in healthy people. For people with various dysfunctions, these parameters may be different),(2)difficulties in locating the sacrococcygeal joint—participants with deformities of the coccyx and sacrum, as well as with a large amount of tissue in this area, were excluded due to the inability to precisely define the measuring point,(3)neuromuscular diseases—they disturb the possibility of precise posture correction and its maintenance during the measurement,(4)participation in specialized postural disorder therapy—such activities involve learning to correct body posture in various ways, which could affect the results achieved by these participants,(5)a diagnosis of scoliosis—this disease causes the occurrence of significant deformations of the skeletal system and the correction of the figure in these cases should be implemented in a very individualized way,(6)previous spinal surgeries and receiving analgesics, sternal and chest deformities—in these people, the shape and mobility of the parts of the body subjected to the procedure may differ significantly from physiology, which could have a negative impact on the obtained results.

The children and their legal guardians provided voluntary written informed consent for participation in the experiment. Information about the procedure was provided, and the objective of the study, safety and privacy matters, including the use of photographs, were explained and the study protocol was presented.

The approval of the Bioethics Committee of the Faculty of Medicine and Health Sciences of Jan Kochanowski University in Kielce no. 17/2016 was obtained for the study.

### 2.2. Testing Method

A planned simultaneous measurement of the following was performed during a single examination session:

I. Angle parameters of the alignment of the body of sternum (α angle) and sacrum (β angle) relative to the body’s sagittal axis, assessed using the Saunders digital inclinometer (Baseline Digital Inclinometer Range of Motion Measurement Tool). The device has a resolution of 0.1 degrees and the measurement accuracy is +/−1 degree. In the α angle study, the Saunders inclinometer was applied to the anterior surface of the sternum body (Figure 1), and in the β angle study, one foot of the inclinometer was placed against the surface of the sacral joint and the other foot was placed against the surface of the medial sacral crest (Figure 2).

In the case of the measurement of the angle of the sternum body, the differences between the measurements performed ten times by the same researcher (intraobserver repeatability) were not statistically significant (*p* = 0.38). SEM is 2.4°.

In the case of the measurement of the angle of the sacrum, the differences between the measurements performed ten times by the same researcher (intraobserver repeatability) were not statistically significant (*p* = 0.24). SEM is 3.7°.

In the case of measuring the angle of the sternum body inclination performed by three interobserver repeatability researchers, the intraclass correlation coefficient (ICC) was 0.86, and the chi2 test did not show significant differences between individual examiners (*p* = 0.54), CI: 0.74/0.91.

In the case of a measurement of the angle of inclination of the sacrum by three investigators, the interobserver repeatability factor, the ICC is 0.90, and the chi2 test shows no significant differences between the individual examiners (*p* = 0.37). CI: 0.79/0.93.

The reliability of the measurements with the Cronbach’s alpha test was high and was at the level of 0.86 for the measurement of the sternal bone shaft angle and 0.91 for the measurement of the sacrum angle.

The high reproducibility of the obtained results allows us to state the reliability of the measurements made with the Saunders inclinometer [30].

II. Angle parameters of the thoracic (kyphosis—ω_1_ angle) and lumbar (lordosis—ω_2_ angle) spine curvatures, assessed using the DIERS Formetric 4D system (DIERS Formetric III 4D model by DIERS INTERNATIONAL GMBH). The resolution of the device is 0.01 degrees and the accuracy is 0.25 degrees.

The position of the patient during the examination is shown in Figure 3.

Based on these, the common sense was calculated for:(a)sacral angle and sternal angle: γ = β − α,(b)sternum and thoracic kyphosis angles: γ_1_ = 180 − (α + ω_1_),(c)sternum and lumbar lordosis angles: γ_2_ = 180 − (α + ω_2_).

The position of the angles used in the work (sternum (α), sacrum (β), thoracic kyphosis (ω_1_) and lumbar lordosis (ω_2_)) is shown schematically in Figure 4. The lines a and b are a reference to the horizontal plane.

During the physical examination, the subject sat on a horizontal seat of variable height, placing equal loads on the ischial tuberosities and with the lower extremities flexed at the hip and knee joints to 90°. The feet were placed flat on the floor, hip-width apart. The upper extremities were loose, with the hands resting on the thighs. To choose the optimal posture according to Mork and Westgard’s protocol [31], the subjects adopted each posture three times. The measurement was carried out once after the subjects assumed the optimal position.

For the α angle assessment, a Saunders inclinometer was placed on the anterior surface of the body of sternum, and for the β angle assessment, one inclinometer foot was placed on the sacrococcygeal joint surface and the other one on the median sacral crest surface. All measurements were performed by the same person trained in the use of the tools used.

Measurements were performed in a sitting position in the following postures:In a passive, relaxed posture without back support or active muscle involvement, with posterior pelvic tilt, referred to as the passive posture.In the corrected, active posture without back support. Under the supervision of the examiner, the subject moved into the corrected posture, assessed as complete active physiological extension of the spine in the easiest manner for the subject, by: lifting the sternum diagonally, increasing forward pelvic tilt, retracting the head with the mandible parallel to the ground, setting the shoulder blades in the “back and down” position (alignment of the shoulder blade in the posterior depression, according to PNF) and a slight forward tilt of the trunk [28]. It was observed that in order to achieve the intended correction, it is necessary to control the alignment of the body of sternum and sacrum both verbally and manually.

### 2.3. Statistical Analysis

A statistical analysis of the results obtained during the study was performed using the StatSoft Statistica 13.1 software. The statistical significance level for the analysis was assumed for *p* < 0.05. The basic descriptive statistics were calculated.

The D’Agostino–Pearson test was used to calculate the normal distribution. It also allows identification of the causes of data distribution disturbances. Most of the presented data showed the features of a normal distribution, therefore parametric tests were used in the analyses.

The significance of the results was tested using the dependent samples *t*-test, the Pearson correlation coefficient (r) and the confidence intervals for the mean results. Stepway regression analysis was used to estimate the impact of the individual outcomes of “common sense.” Confidence intervals were determined for the common sense [30]: γ between the body of sternum α and sacrum β, γ_1_—between the body of sternum α and thoracic kyphosis ω_1_ and γ_2_—between the sacrum β and lumbar lordosis ω_2_.

## 3. Results

### 3.1. Results of Measurements of Body of Sternum (α), Sacrum (β), Thoracic Kyphosis (ω_1_) and Lumbar Lordosis (ω_2_) Angles

A summary of results for females and males is shown in Table 1. Particular attention is paid to much larger standard deviations of individual parameters in the passive position compared to the corrected position (Table 1).

### 3.2. Percentage Errors

Percentage errors (E_p_%) for the common senses were estimated based on the descriptive statistics provided in Table 1. The differences between the theoretical (M_e,t_) and empirical (M_e,emp_) medians shown in Table 2 are much greater for the passive posture (Table 2).

A lack of regularity, i.e., the complementary alignment of body parts to achieve the desired effective corrected sitting posture, was identified in both males and females in the passive posture. This is reflected in the differences in errors between the empirical and theoretical medians for the common senses γ, γ_1_, γ_2_. This regularity, or harmony, was found for the corrected posture in males and females.

### 3.3. Differences between the Corrected and Passive Postures

The results for the corrected and passive postures were compared. For this purpose, the sample was divided into two groups according to sex and dependent samples *t*-tests were performed (Table 3).

The results in Table 3 indicate that the common sense γ_1_ is greater for the corrected posture, while in other cases it is greater for the passive posture.

A comparison of the corrected and passive postures in females is provided in Table 3. All differences are statistically significant and only the common sense γ_1_ is greater in the passive posture than in the corrected posture (Table 3).

### 3.4. Differences between Females and Males

The parameters were analyzed for differences in individual angles within a given sex. The results of these analyses are provided in Table 3. The only significant difference was found for the angle of kyphosis, which is greater in males than in females in the passive posture (Table 3).

### 3.5. Relationship between Angles

The relationship between angles was analyzed. The Pearson’s r test was used for this purpose. Table 4 contains the common sense results for the sternum and sacrum in the corrected posture; it is negatively correlated with the sternal angle both in males and females, and positively correlated with the sacral angle. The common sense of the sternum and kyphosis alignment (γ_1_) in the corrected posture was negatively correlated with the sternal slope and kyphosis. The common sense for the sacrum and lordosis (γ_2_) was negatively correlated with lordosis in both sexes but positively correlated with the sacral slope (Table 4). In the passive posture, the relationships are different. The common sense for the sternum and the sacrum is negatively correlated with the sternal slope angle in both sexes, whereas it is positively correlated with the sacral slope angle in females only. The common sense for the alignment of sternum and kyphosis in males was negatively correlated with the sternal slope and kyphosis angles, whereas in females it was only negatively correlated with kyphosis. The common sense of lordosis and sacrum was only negatively correlated with lordosis in females (Table 4).

Table 4 contains the correlations existing between all of the described angles in the corrected posture in both sexes. In the passive posture, these relationships are clearly disturbed and in many cases are not statistically significant.

In order to estimate the influence of individual variables on the assessed angles, a stepway regression analysis was performed (Table 5). The model turned out to be important. Most of the proposed predictors had a significant impact on the size of the measured angles. The exception is γ_2_ which does not significantly affect the size of the sacrum angle in a passive position neither in males nor in females, as originally assumed.

Detailed analyses showed a significant influence of the angle of the sternum and sacrum on the parameters of posture in the sagittal plane. In the passive position, the angle of the sternum influenced the kyphosis angle (0.69) and the angle of the sacrum on the size of the lumbar lordosis (0.51). In the corrected position, the influence of the size of the position of the sacrum on the size of the lumbar lordosis was higher (0.77).

### 3.6. Confidence Intervals

Using the relevant formulas [30], the confidence intervals for the measured angles were calculated in males and females (Table 6). There are substantial differences in results between the corrected and passive postures. The confidence intervals for all angles in the corrected posture are characterized by a significantly smaller range, which indicates a greater measurement homogeneity. Moreover, the confidence intervals for the thoracic kyphosis and lumbar lordosis angles in the corrected posture are similar to the reference values described in the literature [32,33,34], which indicates the clinical value of body alignment in this posture.

## 4. Discussion

The markers of alignment in the corrected sitting body posture proposed by the “common sense” are characterized by homogeneous results (Table 1). These parameters have been developed for a homogenous population of healthy subjects. It was observed that in order to achieve the intended correction, it is necessary to control the alignment of the body of sternum and sacrum both verbally and manually.

In the study setting, a high measurement repeatability was observed when determining the corrected body posture, including for the body of sternum and sacral angles (Table 2). Moreover, the proposed corrected sitting posture includes the assumptions of the proper spatial body alignment. Extensive clinical experience shows that even people with almost identical physiology use different mechanisms of maintaining the vertical position (correct muscle correction, pathological muscle correction, use of passive elements of the musculoskeletal system, etc.), which results in a different end result of these actions. In this case, the homogeneous results obtained in the corrected position indicate the effectiveness of the proposed method of posture correction. The correct selection of predictors in the form of “common meanings” for the relationship of individual angles to each other is confirmed by regression analyses (Table 5). Such an approach to the analysis allows a slightly broader look at the mutual relations of individual parameters describing body posture. The angle of kyphosis depends on the position of the sternum, and the angle of lordosis depends on the slope of the sacrum. If these two values significantly determine the curvature of the spine in the sagittal plane, then they can be combined to generate a single variable “common sense.” This allows description of the degree of correct body posture in the sagittal plane with one parameter.

It should be noted that it is considered easy to assume by most subjects [28]. This statement encourages correcting body posture with no additional equipment, such as special chairs. The justification for this conclusion is provided by Vergara and Page, who point out that mild discomfort while writing in a sitting position was not significantly different between using a standard office chair without a backrest and the Back App corrective chair [35,36]. Therefore, the concept of assuming the corrected sitting posture should include implementing interventions to reduce the effort connected with sitting properly [37].

When observing the change in sternal alignment, a correction of the entire posture, including the alignment of the head and pelvis, may be noted [38,39]. It is assumed that a harmonious and balanced alignment will be created, in which local and global muscles activated by reflex should protect the spine from strain when performing daily activities [40,41].

In practical management, the person of interest’s attention is distracted from their determination to “sit straight” and is instead focused on a specific task. The task is for the subject of corrective actions to concentrate on slight—from the subject’s perspective—lifting of the body of sternum to an angle of approximately 62–66.5° (Table 6) relative to the body’s sagittal axis. This angle is the optimal alignment for both males and females (Table 3).

Additionally, pelvic retraction is implemented, along with a slight forward tilt of the trunk if there is no spinal support [28,37,40] and a slight backward tilt if the spine is supported. Finally, the physiological alignment of the lumbar lordosis, thoracic kyphosis and pelvis [32] is attainable for both sexes in the corrected posture, as demonstrated by the presented results (Table 4), with the alignment of the body of sternum at an angle of approx. 64° (Table 3).

However, at the initial stage of developing the proper body posture habit, there may be difficulties in determining or recreating it by oneself [19]. For subjects with locomotor system disorders, the corrective actions may be difficult to achieve at the initial stage. However, clinical practice and research by other authors show that removing local disorders ultimately allows achieving or coming close to the ideal posture [42].

It should be noted that eliminating improper habitual behaviors is associated with a discomfort that decreases in young people only after 3–4 months of systematic work [43]. Early postural evaluation may help prevent or at least reduce spinal problems at later stages of life [44]. Failure to propagate the corrective actions consolidates changes that may have substantial implications for general physical fitness at a mature age [45,46].

The review of collected literature indicates that the papers on sagittal alignment evaluation mostly examined the pelvic alignment angles: pelvic incidence and pelvic tilt, as well as sacral slope and lumbar lordosis angle in relation to various parameters being assessed, i.e., C7 plumbline slope angle, thoracic kyphosis angle, thoracic tilt, lumbar tilt, sagittal vertical axis [34,47,48,49,50,51,52].

The results for parameters measured in the papers referenced above are presented in the original manner. Therefore, pelvic incidence, pelvic tilt and sacral slope were referenced to the lumbar lordosis angle [50], to the lumbar lordosis angle along with hip joint alignment [50] and finally to the lumbar lordosis angle along with the C7 plumbline slope angle [38], demonstrating the existing relationships between the investigated parameters. The individual authors presented very interesting results of a correlation matrix between the examined parameters. Mac-Thiong et al. were searching for common relationships for the pelvic tilt and sacral slope, thoracic kyphosis angle, lumbar lordosis angle, thoracic and lumbar spine slope angle [51]. Asai et al. were searching for common relationships for the body’s vertical axis slope angle, thoracic kyphosis angle, pelvic incidence and lumbar lordosis angle [49]. Ghandhari et al. were searching for common relationships for the thoracic kyphosis angle, lumbar lordosis angle, thoracic spine slope angle, lumbar spine slope angle, pelvic incidence, pelvic tilt, sacral slope and body’s vertical axis [34].

The results in the above papers present the importance of the investigated parameters for estimating body posture through different forms of interpretation. The results in these papers clearly indicate the statistically significant correlation and simultaneity of movements of the pelvis, individual parts of the spine and the body’s axis.

The cited studies indicate that other researchers also see the need to define the mutual relations of individual parts of the body in order to define body posture. In the light of the research carried out and the literature cited, the development of this concept of body posture assessment seems to be a good direction for future analyses.

In our own clinical observations, the usefulness of using the α angle in the position of about 64 degrees in the treatment of various dysfunctions in people of different ages has been noticed. It should be emphasized that the correction of the figure depends on the lack of structural changes in the musculoskeletal system. The presented results apply to a population of healthy people. The authors recognize the need to define similar indications for the population suffering from various dysfunctions, in particular spinal deformities and low back pain syndromes. People with these ailments were not qualified for the presented study, which may be its limitation.

## 5. Conclusions

Changes in the alignment of the body of sternum and sacrum resulted in changes in the thoracic kyphosis and lumbar lordosis angle values, which may be an important component of clinical observations of sitting body posture in children.Implementing the body of sternum alignment angle of about 64° relative to the body’s sagittal axis in clinical practice as one of the objectives of postural education may be the target solution for sitting body posture correction in children.

## Figures and Tables

**Figure 1 ijerph-19-16287-f001:**
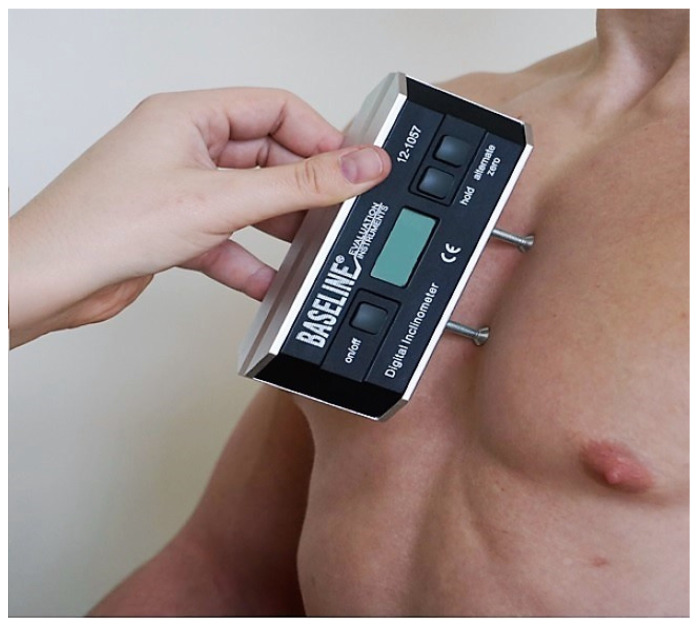
Measurement of the angle of the body of the sternum.

**Figure 2 ijerph-19-16287-f002:**
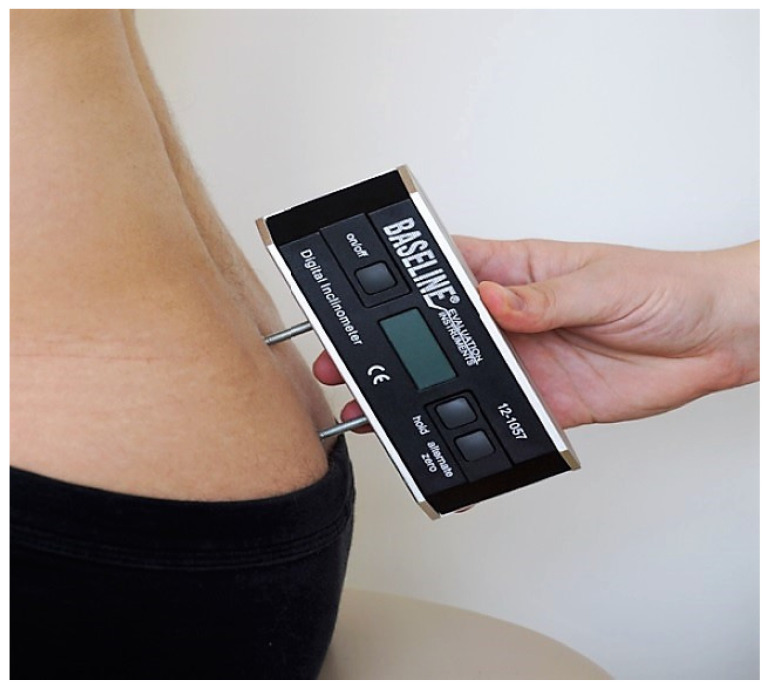
Sacrum angle measurement.

**Figure 3 ijerph-19-16287-f003:**
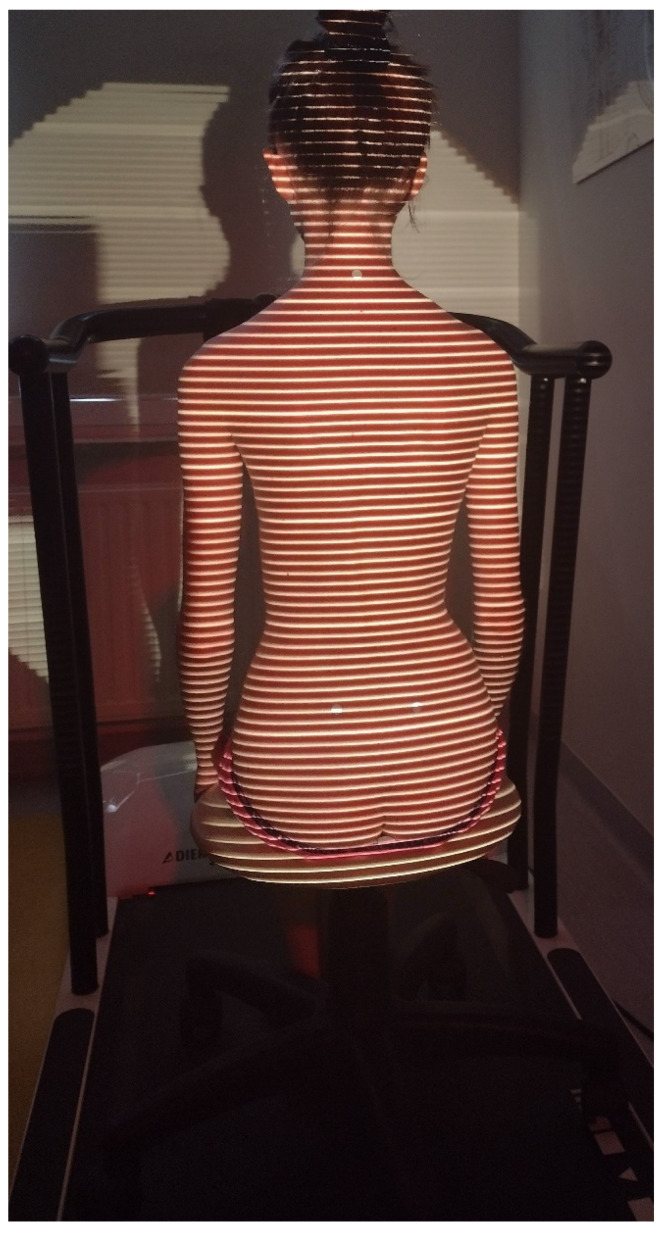
DIERS Formetric Measurement.

**Figure 4 ijerph-19-16287-f004:**
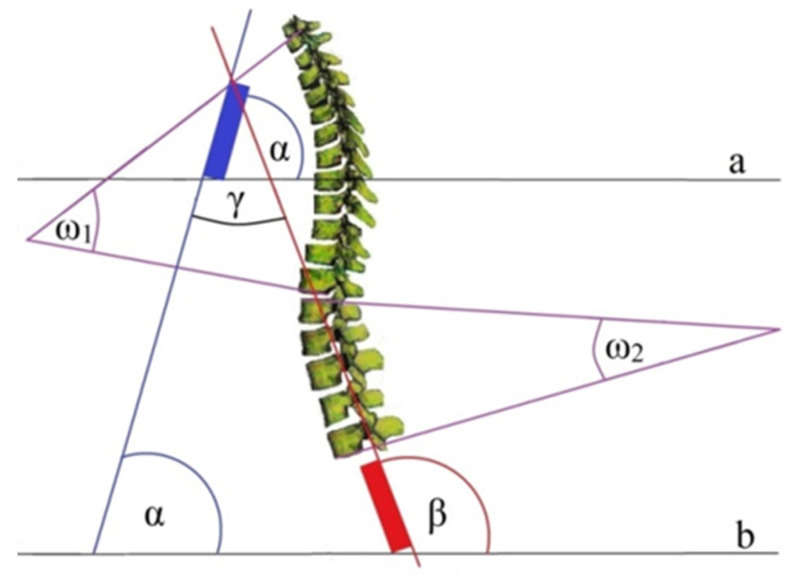
The common system of the position of the sternal body and the sacrum in relation to thoracic kyphosis and lumbar lordosis [30].

**Table 1 ijerph-19-16287-t001:** Basic computational statistics results.

Posture	Angle [⸰]	Males				Females			
		M	SD	Me	Mo	M	SD	Me	Mo
Corrected	α	64.47	3.5	64.35	65	64.46	4.5	65	65.0 ^1^
	β	112.65	1.85	112	112	112.24	2.99	112.15	108.0 ^1^
	ω_1_	38.11	6.48	38.06	26.43 ^1^	39.21	5.24	38.2	31.11 ^1^
	ω_2_	38.42	6.1	37.08	31.54 ^1^	38.82	5.87	37.55	30.57 ^1^
	γ	48.13	3.02	48.1	47	46.83	4.08	45.2	45
	γ_1_	77.42	8.15	78.63	60.05 ^1^	76.33	6.83	75.61	63.17 ^1^
	γ_2_	77.11	8.21	79.32	60.77 ^1^	76.72	6.35	76.68	61.96 ^1^
Passive	α	94.54	16.63	96.2	108	90.02	11.6	92	92.7
	β	76.7	9.76	76.25	85	66.82	15.19	67.7	72
	ω_1_	58.57	15.68	62.74	23.75 ^1^	49.07	11.96	48.28	25.11 ^1^
	ω_2_	−18.92	7.27	−18.52	−31.90 ^1^	−15.69	8.13	−15.98	−26.61 ^1^
	γ	−17.84	17.62	−21	−23	−11.03	11.18	−12.5	−14
	γ_1_	26.89	29.09	22.51	−22.07 ^1^	40.91	17.25	38.08	6.75 ^1^
	γ_2_	104.38	18.27	103.03	60.70 ^1^	105.67	12.83	105.54	74.27 ^1^

M—mean, SD—standard deviation, Me—median, Mo—modal. ^1^ There is more than one mode.

**Table 2 ijerph-19-16287-t002:** Difference between the corrected and passive postures.

Common Sense/Angle		Corrected Posture			Passive Posture		
		M_e,t_	M_e,emp_	E_p_%	M_e,t_	M_e,emp_	E_p_%
Males	γ	48.1	48.95	1.74	−17.84	−16.33	28.53
	γ_1_	89	78.63	13.19	30.10	75.61	60.19
	γ_2_	88.00	79.32	10.94	133.5	76.68	74.10
Females	γ	45.2	48.2	6.24	−12.5	−8.82	41.79
	γ_1_	83.68	75.61	10.67	85.10	22.51	278.07
	γ_2_	86.56	76.68	12.88	126.60	105.54	19.96

M_e,t_—theoretical median, M_e,emp_—empirical median, E_p_%—percentage errors.

**Table 3 ijerph-19-16287-t003:** Differences between the corrected and passive posture and comparison of measurements.

Posture/Angle		Differences between the Corrected and Passive Postures				Comparison of Measurements
		Males		Females		
		Mean (⸰)	*p*	Mean (⸰)	*p*	*p*
α	Corrected	64.47	<0.001	64.46	<0.001	>0.05
	Passive	94.54		90.02		>0.05
β	Corrected	112.65	<0.001	112.24	<0.001	>0.05
	Passive	76.7		79.27		>0.05
ω_1_	Corrected	38.11	<0.001	39.21	<0.001	>0.05
	Passive	58.57		49.07		<0.05
ω_2_	Corrected	38.40	<0.001	38.82	<0.001	>0.05
	Passive	−18.92		−15.69		>0.05
γ	Corrected	48.13	<0.001	46.83	<0.001	>0.05
	Passive	−17.84		−11.04		>0.05
γ_1_	Corrected	77.42	<0.001	76.33	<0.001	>0.05
	Passive	26.89		40.91		>0.05
γ_2_	Corrected	77.11	<0.001	76.72	<0.001	>0.05
	Passive	104.38		105.67		>0.05

**Table 4 ijerph-19-16287-t004:** Correlations between angles.

Posture	Common Sense	Angle	Males		Females	
			r	*p*	r	*p*
Corrected	γ	α	−0.77	<0.001	−0.48	<0.05
		β	0.75	<0.001	0.93	<0.01
	γ_1_	α	−0.64	<0.01	−0.64	<0.01
		ω_1_	−0.91	<0.001	−0.75	<0.001
	γ_2_	β	0.17	<0.05	0.46	<0.01
		ω_2_	−0.92	<0.001	−0.73	<0.001
Passive	γ	α	−0.64	<0.01	−0.73	<0.001
		β	0.82	<0.01	0.71	<0.01
	γ_1_	α	−0.91	<0.001	−0.25	>0.05
		ω_1_	−0.89	<0.001	−0.74	<0.001
	γ_2_	β	−0.33	>0.05	0.33	>0.05
		ω_2_	−0.42	>0.05	−0.46	<0.05

r—Pearson’s r test score.

**Table 5 ijerph-19-16287-t005:** Results of the regression analysis of individual “common senses” in relation to the analyzed angles.

Posture	Common Sense	Angle	Males		Females	
			b	*p*	b	*p*
Corrected	γ	α	−0.66	<0.001	−0.93	<0.001
		β	0.64	<0.001	0.86	<0.001
	γ_1_	α	−0.79	<0.001	−0.66	<0.001
		ω_1_	−0.43	<0.001	−0.77	<0.001
	γ_2_	β	0.17	<0.05	0.46	<0.05
		ω_2_	−0.92	<0.001	−0.73	<0.001
Passive	γ	α	−0,72	<0.01	−0.72	<0.001
		β	0.86	<0.001	0.68	<0.001
	γ_1_	α	0,57	<0.001	−0.69	<0.001
		ω_1_	0,54	<0.001	−0.67	<0.001
	γ_2_	β	−0.42	>0.05	0.33	>0.05
		ω_2_	−0.49	<0.05	−0.46	<0.05

b—the b coefficient of individual predictors.

**Table 6 ijerph-19-16287-t006:** Confidence intervals for angles measured.

Posture/ Angle	Males		Females	
	Corrected	Passive	Corrected	Passive
α	62.85–66.08	86.86–102.22	62.38–66.54	84.66–95.38
β	111.17–114.13	69.75–83.65	110.66–113.81	74.51–84.04
ω_1_	35.12–41.10	51.33–65.81	36.79–41.63	43.54–54.60
ω_2_	35.60–41.23	−22.28–15.56	36.11–41.53	−19.45–11.93
γ	46.14–50.12	−30.79–−4.89	44.65–49.02	−17.52–−4.56
γ_1_	73.65–81.19	13.45–40.33	73.17–79.49	32.94–48.88
γ_2_	73.32–80.90	95.94–112.82	73.78–79.65	99.74–111.60

## Data Availability

The raw data supporting the conclusions of this article will be made available by the authors, without undue reservation.

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
