# Peer review of "Alignment of the Sternum and Sacrum as a Marker of Sitting Body Posture in Children"

_ijerph, 2022, doi:10.3390/ijerph192316287_

Round 1

Reviewer 1 Report (Previous Reviewer 1)

Thank you for improving the manuscript.

Reviewer 2 Report (Previous Reviewer 2)

I appreciate the authors addressing my concerns. 

This manuscript is a resubmission of an earlier submission. The following is a list of the peer review reports and author responses from that submission.

Round 1

Reviewer 1 Report

Some corrections should be made to improve the quality of this interesting manuscript. It is written in a very chaotic way and carelessly:

1. Introduction should be extended to support the aim of this study. Objectives should be more precisely defined and/or hypotheses should be stated.

2. Material and methods:

- Inclusion and exclusion criteria - please write whole sentences, not only nouns; the same about testing method;

- How the structure of the chest/spine was assessed?

- "spinal pain" - how long before the study?

- "A summary of results for females and males is shown in Table 1." - this should be in results.

- "Subjects were enrolled into the study randomly." - this should be in the description of subjects. Additionally, it should be better defined how this randomisation looked like.

- What was the resolution of the angle measurement? What is the producer's name of the inclinometer/Diers system? All details allowing to repeat the study protocol must be defined!

- Please provide figures presenting measurement method in a described sitting position;

- How the normality of data distribution was calculated?

- What was the reason of calculating CI? CI-values alone don't give significant information (more than calculated anyway).

- Due to defined aim of this study, regression coefficients should be calculated;

3. Results are written in completely unacceptable way - the same values are repeated in tables; shortcuts are not explained below the tables; too many values are in tables (for parametrical tests there is no reason to put there median, mode and min-max; t-test values are not, in my opinion, obligatory); each table should be mentioned in text BEFORE; significant values in tables should be in bold or italic - the reader have to see what the authors want to show.

4. Sentences like this: "This confirms that the arithmetic mean and median have similar values for the corrected posture results." are interpretation of the results and should not be in "Results" section.

5. Discussion is rather poor and should be extended.

6. Minor comments to discussion:

- lines 226-228: homogenous results from homogenous group of patients in nothing suprising and do not support the thesis, that the measurement method is the right one;

- "It was observed that in order to achieve the intended correction, it is necessary to control the alignment of the body of sternum and sacrum both verbally and manually." this information should be included in methods and results. Informal result should not be included in results interpretation.

Author Response

Thank you very much for all valuable comments. We have made every effort to fulfill all suggestions. All changes made to the manuscript are visible in the change tracking option in the attached DOCX file. We hope that the current version will meet your expectations. Below, we will address all of the remarks in turn:

Point 1: Introduction should be extended to support the aim of this study. Objectives should be more precisely defined and/or hypotheses should be stated.

Response: Thank you very much for the suggestion. The introduction has been extended to include this point.

Point 2: Material and methods:

- Inclusion and exclusion criteria - please write whole sentences, not only nouns; the same about testing method;

Response: These points have been extended as suggested.

- How the structure of the chest/spine was assessed?

Response: In the qualification process, it was done by a qualified physiotherapist with extensive experience. This information was included in the Inclusion criteria section.

- "spinal pain" - how long before the study?

Response: Inclusion criteria included no discomfort during the measurements and within three months this information was added in the relevant section.

- "A summary of results for females and males is shown in Table 1." - this should be in results.

Response: Thank you for your suggestion. This has been corrected.

- "Subjects were enrolled into the study randomly." - this should be in the description of subjects. Additionally, it should be better defined how this randomisation looked like.

Response: Subjects were enrolled into the study randomly. For the experiment, participants were recruited from among children presenting for routine body posture testing. All children meeting the inclusion criteria were selected consecutively until the presented number of qualified participants was reached. Participants were reported for check-ups by their guardians using the IT system. This information has been added to the appropriate section.

- What was the resolution of the angle measurement? What is the producer's name of the inclinometer/Diers system? All details allowing to repeat the study protocol must be defined!

Response: Thank you for this suggestion. Missing information has been completed in the appropriate section.

- Please provide figures presenting measurement method in a described sitting position;

Response: Thank you for the suggestion. The text has been supplemented with relevant figures.

- How the normality of data distribution was calculated?

Response : The D'Agostino-Pearson test was used to calculate the normal distribution. It also allows to identify the causes of data distribution disturbances. Most of the presented data showed the features of a normal distribution, therefore parametric tests were used in the analyzes. The Statistica software used for the analysis gives the possibility of adjusting the distribution, which, with the obtained D'Agostino-Pearson test values, does not significantly disturb the results.

- What was the reason of calculating CI? CI-values alone don't give significant information (more than calculated anyway).

Response: Confidence intervals were calculated to visualize the variety of passive positions taken by the test subjects and to determine the homogeneity of the corrected position. The authors are aware that this is not a precise result, but it allows us to answer the question posed in the introduction: "is there one correct sitting position in healthy people?"

- Due to defined aim of this study, regression coefficients should be calculated;

Response : Thank you very much for this suggestion. In the submitted work, the concept of "common sense" was used to analyze the research material, which allows for the generalization of the research results. This generalization occurs when the H0 hypothesis is rejected for the test subjects. Regression also serves as a generalization. The authors are aware that regression is a more popular statistical method for this type of analysis, but in this paper we want to maintain this style of data processing. Regression will certainly be used in our next studies.

Point 3: Results are written in completely unacceptable way - the same values are repeated in tables; shortcuts are not explained below the tables; too many values are in tables (for parametrical tests there is no reason to put there median, mode and min-max; t-test values are not, in my opinion, obligatory); each table should be mentioned in text BEFORE; significant values in tables should be in bold or italic - the reader have to see what the authors want to show.

Response : Thank you very much for your attention. The tables have been improved: the amount of data in the table has been reduced, the most important have been bolded, a legend has been added with the description of abbreviations under the tables. All tables are cited in the text that precedes them.

Point 4: Sentences like this: "This confirms that the arithmetic mean and median have similar values for the corrected posture results." are interpretation of the results and should not be in "Results" section.

Response : Thank you for your good attention. Phrases relating to the interpretation of results have been removed from this section.

Point 5: Discussion is rather poor and should be extended.

Response : Thank you for this suggestion. The Discussion section has been reorganized and supplemented.

Point 6: Minor comments to discussion:

- lines 226-228: homogenous results from homogenous group of patients in nothing suprising and do not support the thesis, that the measurement method is the right one;

Response : Thank you for this suggestion. The indicated results do not support the value of the measurement method, this is a fair remark. Extensive clinical experience shows that even people with almost identical physiology use different mechanisms of maintaining the vertical position (correct muscle correction, pathological muscle correction, use of passive elements of the musculoskeletal system, etc.), which results in a different end result of these actions. In this case, the homogeneous results obtained in the corrected position indicate the effectiveness of the proposed method of posture correction. This explanation is included in the Discussion section.

- "It was observed that in order to achieve the intended correction, it is necessary to control the alignment of the body of sternum and sacrum both verbally and manually." this information should be included in methods and results. Informal result should not be included in results interpretation.

Response : Thanks for your suggestion. This information is included in the Method section.

Reviewer 2 Report

Overall I believe this manuscript has merit however, there are several things that need to be addressed. One of my biggest concerns is the organization and writing. Please have someone thoroughly proof-read this manuscript as many sentences seem to be run-ons (even though grammatical structure is correct) and the paragraphs are either really long or a few sentences. 

Introduction

While the introduction cites several studies about the psychophysical aspect of sitting posture, it fails to make a case for why there is a need to examine these posture in children. Prior to the objective statement the authors should add additional information about posture in children, specifically pre-adolescents as these are the subjects used in this study.

Methodology

There are significant gaps in the methodology that make it hard to replicate

1. How were participants recruited?

2. Did you use the digital incliometer? 

3. Did the same researcher perform all the measurements? 

4. What the inter- and/or intra-rater reliability on these measures? 

5. You should put some of the information of how many times participants were measured in the section describing the instrument?

6. What do you mean by subjects were enrolled in this study randomly?

7. Were all participants measured by the same tester? Different testers? What was done to control for reliability of data?

8. Prior to performing the analyses were data tested for normality? If so were data normally distributed? If not, what techniques were used to normalize data.

9. Considering the large 95% CIs I am very curious about your data distribution.

Results

You can combine Tables 3 and 4

Although I do not work in pediatric, I know that these angles change with age (my work is 18+). Is this not the case with young people? If it is not the case please provide that information in your introduction. However, if there are changes in angles with age, please control for age in your analysis.

You can put the 95% CI information with the mean difference information. 

Discussion

The first paragraph in the discussion is VERY confusing. You should use this paragraph to summarize the major results of your findings based on the objectives of your study. 

The rest of the discussion makes sense based on the data however, I would recommend breaking up the really long paragraphs. 

Author Response

Thank you very much for all valuable comments. We have made every effort to fulfill all suggestions. All changes made to the manuscript are visible in the change tracking option in the attached DOCX file. We hope that the current version will meet your expectations. Below, we will address all of the remarks in turn:

Point 1: Overall I believe this manuscript has merit however, there are several things that need to be addressed. One of my biggest concerns is the organization and writing. Please have someone thoroughly proof-read this manuscript as many sentences seem to be run-ons (even though grammatical structure is correct) and the paragraphs are either really long or a few sentences. 

Response : Thank you very much for this suggestion. Paragraphs and individual sentences have been checked and in many cases broken down as suggested.

Point 2: Introduction

While the introduction cites several studies about the psychophysical aspect of sitting posture, it fails to make a case for why there is a need to examine these posture in children. Prior to the objective statement the authors should add additional information about posture in children, specifically pre-adolescents as these are the subjects used in this study.

Response : Thank you very much for this suggestion. The introduction has been supplemented with relevant content.

Point 3: Methodology

There are significant gaps in the methodology that make it hard to replicate

  1. How were participants recruited?

Response : Subjects were enrolled into the study randomly. For the experiment, participants were recruited from among children presenting for routine body posture testing. All children meeting the inclusion criteria were selected consecutively until the presented number of qualified participants was reached. Participants were reported for check-ups by their guardians using the IT system. This information has been added in the appropriate section of the text.

  1. Did you use the digital incliometer? 

Response : Yes, the study used a Saunders digital inclinometer (Baseline Digital Inclinometer Range of Measurement Tool). This information as well as the frequency and accuracy of the measurement have been completed in the Method section.

  1. Did the same researcher perform all the measurements? 

Response : Yes, all measurements were made by the same trained person. He was a physiotherapist with great clinical experience, professionally handling both research tools. The method section has been supplemented with this information.

  1. What the inter- and/or intra-rater reliability on these measures? 

Response : In the case of the measurement of the angle of the sternum body, the differences between the measurements performed ten times by the same researcher (intraobserver repeatability) were not statistically significant (p = 0.38). SEM is 2.4⁰.

In the case of the measurement of the angle of the sacrum, the differences between the measurements performed ten times by the same researcher (intraobserver repeatability) were not statistically significant (p = 0.24). SEM is 3.7 °.

In the case of measuring the angle of the sternum body inclination performed by three interobserver repeatability researchers, the intraclass correlation coefficient (ICC) was 0.86, and the chi2 test did not show significant differences between individual examiners (p = 0.54), CI: 0.74 / 0 , 91.

In the case of a measurement of the angle of inclination of the sacrum by three investigators, the interobserver repeatability factor

The ICC is 0.90, and the chi2 test shows no significant differences between the individual examiners (p = 0.37). CI: 0.79 / 0.93.

The reliability of the measurements with the Cronbach's Alpha test was high and was at the level of 0.86 for the measurement of the sternal bone shaft angle and 0.91 for the measurement of the sacrum angle.

The high reproducibility of the obtained results allows us to state the reliability of the measurements made with the Saunders inclinometer.

These analyzes are performed for other work using a Saunders digital inclinometer. These results were not cited in the submitted work, as they are available in several studies. The authors did not want to introduce additional variables, so this issue was omitted in the description of the tool.

  1. You should put some of the information of how many times participants were measured in the section describing the instrument?

Response : The text reads: "To choose the optimal posture according to Mork and Westgard's protocol [16], the subjects adopted each posture three times." According to the suggestion, this information has been supplemented with the following information: "The measurement was carried out once after the subjects assumed the optimal position".

  1. What do you mean by subjects were enrolled in this study randomly?

Response : Thank you very much for drawing your attention to this issue. Subjects were enrolled into the study randomly. For the experiment, participants were recruited from among children presenting for routine body posture testing. All children meeting the inclusion criteria were selected consecutively until the presented number of qualified participants was reached. Participants were reported for check-ups by their guardians using the IT system. The text has been supplemented with this information.

  1. Were all participants measured by the same tester? Different testers? What was done to control for reliability of data?

Response : All participants were measured by the same tester. The person conducting the measurements has extensive clinical experience and is well trained in the use of both research tools. The research was carried out in accordance with a uniform, detailed protocol. The repeatability and reproducibility of the measurement was carried out before starting the tests and is at a high level. The author of the concept did not participate in data collection, nor did the authors of the methodology.

  1. Prior to performing the analyses were data tested for normality? If so were data normally distributed? If not, what techniques were used to normalize data.

Response : The D'Agostino-Pearson test was used to calculate the normal distribution. It also allows to identify the causes of data distribution disturbances. Most of the presented data showed the features of a normal distribution, therefore parametric tests were used in the analyzes. The Statistica software used for the analysis gives the possibility of adjusting the distribution, which, with the obtained D'Agostino-Pearson test values, does not significantly disturb the results.

  1. Considering the large 95% CIs I am very curious about your data distribution.

Response : Analyzes were performed for the entire population (male and female combined). The results are presented in the table below. As expected, the Confidence intervals for the mean are tighter, but the deviating differences show the confidence intervals for the standard deviation. Similar information is provided in the tables 1 (standard deviations), 7 and 8 (confidence intervals). If this is not necessary, we would prefer to keep the current layout in order not to introduce additional variables and tables.

Variable

 CI -95%

CI 95%

 confidence dev. std. -95%

 confidence dev. std. 95%

 α passive

78,1128

80,8490

7,81291

9,76106

 α corrected

63,3446

64,0228

1,93650

2,41937

ω1 passive

59,0882

63,0337

11,26610

14,07529

ω1 corrected

42,8979

43,9377

2,96899

3,70930

β passive

82,0474

85,1893

8,97118

11,20814

β corrected

112,3929

114,1587

5,04228

6,29957

ω2 passive

-1,9403

3,1626

14,57083

18,20406

ω2 corrected

37,6805

39,0481

3,90511

4,87885

Point 4: Results.

You can combine Tables 3 and 4

Response : Thank you very much for this suggestion, but the second reviewer strongly pointed to the need to reduce the data contained in individual tables. For this reason, the authors decided to leave the division into two separate tables for males and females. If this issue is very important, we will try to discuss it with the other reviewer and find a compromise.

Although I do not work in pediatric, I know that these angles change with age (my work is 18+). Is this not the case with young people? If it is not the case please provide that information in your introduction. However, if there are changes in angles with age, please control for age in your analysis.

Response : Thoracic kyphosis takes its shape after the child is 7 years old. Lordosis at this time is already pre-established, it may still undergo some slight changes during puberty. The authors decided to conduct a study in children of this age to show the biomechanics of the child's body before the changes in the pubertal period.

You can put the 95% CI information with the mean difference information. 

Response : Similar information is provided in the tables 1 (standard deviations), 7 and 8 (confidence intervals). If this is not necessary, we would prefer to keep the current layout in order not to introduce additional variables and tables. Relevant information has been included in the section describing the test group.

Point 5: Discussion

The first paragraph in the discussion is VERY confusing. You should use this paragraph to summarize the major results of your findings based on the objectives of your study. 

Response : Thank you for this suggestion. The Discussion section has been reorganized and supplemented.

The rest of the discussion makes sense based on the data however, I would recommend breaking up the really long paragraphs. 

Response : Thank you very much for this suggestion. Paragraphs and individual sentences have been checked and in many cases broken down as suggested.

Round 2

Reviewer 1 Report

Dear Authors,

thank you for making corrections to the manuscript. Still, it requires many changes:

Major comments:

- the language style still looks like translated in a web-translator; English style should be improved;

- it is good to read some articles, that are already published in this Journal or even Special Issue to learn the way/language, that methods or results are described and to see how to construct the tables and organize the text of the manuscript;

- calculating correlations' coefficients for angles that are used to calculate described indices makes only a little sense; I strongly recommend to calculate regression coeficients using even a simple model like the method of least squares - this would give more information relevant for this study and clinical usage of the described method;

- Introduction section and used references still do not support the aim of this study; the aim/hypotheses are not clearly described; 

- Discussion still can be improved (see "minor comments"), although changes already made, improved this section a lot.

Minor comments:

- inclusion and exclusion criteria should be written like this: "Inclusion criteria included (1)...., (2)..., and (3)..."

- this should not be in Methods: "A summary of results for females and males is shown in Table 1."

- tables 3, 4 &5 - you can write all data in one table, so, that both inter- and intra-group changes are visible;

- tables 7 and 8 should be written as one;

- "In this case, the homogeneous results obtained in the corrected position indicate the effectiveness of the proposed method of posture correction." - but it was not the aim of this study! Probably it is needed to define the aim of the study more precisely or reorganize Discussion section, so that it answers the main question firstly.

Author Response

Major comments:

- the language style still looks like translated in a web-translator; English style should be improved;

Response: This is an objection that is difficult to accept for us. He did not appear in the first version of the review (attention was drawn to the long sentences that were divided). The translation was prepared by a professional company. In the edition of the currently submitted version, we asked for help in editing the text of a native Englishman. We hope that the changes made are now acceptable.

- it is good to read some articles, that are already published in this Journal or even Special Issue to learn the way/language, that methods or results are described and to see how to construct the tables and organize the text of the manuscript;

- calculating correlations' coefficients for angles that are used to calculate described indices makes only a little sense; I strongly recommend to calculate regression coeficients using even a simple model like the method of least squares - this would give more information relevant for this study and clinical usage of the described method;

Response: This has been corrected as recommended.

- Introduction section and used references still do not support the aim of this study; the aim/hypotheses are not clearly described; 

Response: The introduction was extended to justify the choice of the study population. The literature supporting the selection of the population at the age of growth was also significantly supplemented.

- Discussion still can be improved (see "minor comments"), although changes already made, improved this section a lot.

Minor comments:

- inclusion and exclusion criteria should be written like this: "Inclusion criteria included (1)...., (2)..., and (3)..."

Response: This has been corrected as recommended.

- this should not be in Methods: "A summary of results for females and males is shown in Table 1."

Response: This has been corrected as recommended.

- tables 3, 4 &5 - you can write all data in one table, so, that both inter- and intra-group changes are visible;

Response: This has been corrected as recommended.

- tables 7 and 8 should be written as one;

Response: This has been corrected as recommended.

- "In this case, the homogeneous results obtained in the corrected position indicate the effectiveness of the proposed method of posture correction." - but it was not the aim of this study! Probably it is needed to define the aim of the study more precisely or reorganize Discussion section, so that it answers the main question firstly.

Response: This text element was inserted at the explicit suggestion of the first reviewer. If its removal is necessary, we will try to find a compromise with the reviewer who recommended including this sentence.

We hope that the manuscript in its present form meets the publication criteria.

Reviewer 2 Report

I'd like to thank the authors for addressing some of my concerns however, there are still significant gaps in this manuscript.

1. The writing still needs significant work. It is a very difficult read

2. The authors did not provide supporting literature in the introduction to justify the use of this study population.

3. The researchers need to provide intra-rater reliability of the measurements. Instead they just wrote in a person trained to use the tools. 

4. The tables formatting needs to be improved

Author Response

Thank you very much for your valuable comments. We tried to fully cover all of them. Below we will deal with them in turn:

I'd like to thank the authors for addressing some of my concerns however, there are still significant gaps in this manuscript.

  1. The writing still needs significant work. It is a very difficult read

Response: This is an objection that is difficult to accept for us. He did not appear in the first version of the review (attention was drawn to the long sentences that were divided). The translation was prepared by a professional company. In the edition of the currently submitted version, we asked for help in editing the text of a native Englishman. We hope that the changes made are now acceptable.

  1. The authors did not provide supporting literature in the introduction to justify the use of this study population.

Response: The introduction was extended to justify the choice of the study population. The literature supporting the selection of the population at the age of growth was also significantly supplemented.

  1. The researchers need to provide intra-rater reliability of the measurements. Instead they just wrote in a person trained to use the tools. 

Response: This has been corrected as recommended.

  1. The tables formatting needs to be improved

Response: This has been corrected as recommended.

We hope that the manuscript in its present form meets the publication criteria.

Round 3

Reviewer 1 Report

Dear Authors,

despite the changes you made, the manuscript still needs to be improved. Some issues, mentioned in the previous review were not corrected. For details see below.

Response: This is an objection that is difficult to accept for us. He did not appear in the first version of the review (attention was drawn to the long sentences that were divided). The translation was prepared by a professional company. In the edition of the currently submitted version, we asked for help in editing the text of a native Englishman. We hope that the changes made are now acceptable.

The first review included point "Moderate English changes required" - please read the reviews carefully.

Comments:

- inclusion and exclusion criteria should be written like this: "Inclusion criteria included (1)...., (2)..., and (3)...";

- thre are no information about the method of regression analysis, that was used;

- regression analysis is mostly done to analyse multiple factors, that could influence one variable and this was the idea to use this form of calculation in this study, but it seems that you used it only to calculate, that independ variables used to calculate dependent variable are significant in the model, which is mostly clear without stitistics;

- regression analysis is practically not discussed.

Author Response

Thank you very much for your comments.

Comments:

- inclusion and exclusion criteria should be written like this: "Inclusion criteria included (1)...., (2)..., and (3)...";

Answer: This has been corrected

- thre are no information about the method of regression analysis, that was used;

Answer: This has been completed

- regression analysis is mostly done to analyse multiple factors, that could influence one variable and this was the idea to use this form of calculation in this study, but it seems that you used it only to calculate, that independ variables used to calculate dependent variable are significant in the model, which is mostly clear without stitistics;

Answer: The results section (regression analysis) has been enriched with the results concerning the influence of other variables on the shape of the spine in the sagittal plane.

- regression analysis is practically not discussed.

Answer: The regression results have been discussed more fully in the discussion.

We hope that the changes made will prove to be sufficient for the manuscript to be accepted in its present form. Thank you very much for helping us improve the manuscript throughout the review process.

Reviewer 2 Report

Thank you very much for addressing my concerns. I'd like to make sure that the author cite studies that support the need for studying this population rather than their clinical observations. 

Author Response

Comments:  I'd like to make sure that the author cite studies that support the need for studying this population rather than their clinical observations. 

Answer: We have listed the papers that indicate the need to study the population presented in the manuscript. This was done by slightly supplementing the introduction section.

We hope that the changes made will prove to be sufficient for the manuscript to be accepted in its present form. Thank you very much for helping us improve the manuscript throughout the review process.